# Hydrazine High-Performance Oxidation and Sensing Using a Copper Oxide Nanosheet Electrocatalyst Prepared via a Foam-Surfactant Dual Template

**DOI:** 10.3390/nano13010129

**Published:** 2022-12-26

**Authors:** Etab M. Almutairi, Mohamed A. Ghanem, Abdulrahman Al-Warthan, Mufsir Kuniyil, Syed F. Adil

**Affiliations:** Chemistry Department, College of Science, King Saud University, Riyadh 11451, Saudi Arabia

**Keywords:** surfactant, foam, dual templates, copper oxide, nanosheets, hydrazine sensing

## Abstract

This work demonstrates hydrazine electro-oxidation and sensing using an ultrathin copper oxide nanosheet (CuO-NS) architecture prepared via a versatile foam-surfactant dual template (FSDT) approach. CuO-NS was synthesised by chemical deposition of the hexagonal surfactant Brij^®^58 liquid crystal template containing dissolved copper ions using hydrogen foam that was concurrently generated by a sodium borohydride reducing agent. The physical characterisations of the CuO-NS showed the formation of a two-dimensional (2D) ultrathin nanosheet architecture of crystalline CuO with a specific surface area of ~39 m^2^/g. The electrochemical CuO-NS oxidation and sensing performance for hydrazine oxidation revealed that the CuO nanosheets had a superior oxidation performance compared with *bare*-CuO, and the reported state-of-the-art catalysts had a high hydrazine sensitivity of 1.47 mA/cm^2^ mM, a low detection limit of 15 μM (S/N = 3), and a linear concentration range of up to 45 mM. Moreover, CuO-NS shows considerable potential for the practical use of hydrazine detection in tap and bottled water samples with a good recovery achieved. Furthermore, the foam-surfactant dual template (FSDT) one-pot synthesis approach could be used to produce a wide range of nanomaterials with various compositions and nanoarchitectures at ambient conditions for boosting the electrochemical catalytic reactions.

## 1. Introduction

According to recent reports, the global production of hydrazine (diamine, N_2_H_4_) and its derivatives reaches up to 200,000 metric tons a year [1], and it is widely used in explosives and fuels [2], particularly rocket fuel [3]. Moreover, recently, hydrazine has been used in an enormous range of industrial applications, such as corrosion inhibitors [4], reducing agents [5], photographic chemicals [6], drug production [7,8], antioxidants [9], and herbicides [10]. However, the spread of hydrazine usage has increased environmental pollution because it is classified as a highly toxic substance [11]. It has been declared carcinogenic as it can damage the brain, liver, and eyes when absorbed from the metabolized drug or the environment [12,13]. In addition, it has been classified as a B2 group environmental pollutant by the Environmental Protection Agency (EPA) and World Health Organisation (WHO) [14]. Therefore, it is necessary to measure the hydrazine concentration in different types of samples, particularly in aqueous systems.

Several analytical methods, such as chromatography [15], fluorescence methods [16], chemiluminescence [17], and electrochemical methods [18], have been used for the detection of hydrazine in various types of samples. Among these methods, electrochemical techniques are considered to be the best because of their simplicity, low-cost fabrication process, and fast detection response, and because they possess a relatively high sensitivity and selectivity [19,20,21]. However, electrochemical methods with conventional substrates usually show a lower sensitivity and selectivity for sample analysis than electrochemical methods based on a modified electrode [22,23]. Therefore, it is necessary to develop more efficient modified electrodes to achieve the detection of low limits, eliminate interferences, and improve the electrochemical sensing results. Hence, transition-metal oxides and hydroxide nanomaterials of cobalt, nickel, and copper have attracted extensive research attention in recent years for the development of an electrochemical hydrazine sensing platform because of advantages such as a high catalytic activity, low cost, environment friendliness, good stability, and ease of manufacturing and developing electrochemical sensors [24,25].

In particular, among the transition metal oxide nanostructures, copper oxide (CuO) is one of the known p-type semiconductor materials with a narrow band gap of 1.2 eV; it offers interesting photovoltaic, photoconductive, and electrocatalytic properties [26,27,28,29,30]. Many new CuO-based nanoarchitectures with a high specific surface area and mesoporosity, various morphologies, and compositions have shown a significant enhancement in sensor [31,32,33] and electrochemical energy reaction [34,35] applications. For example, CuO nanoparticles [36], nanowires [37], nanoflakes [38], nanoflowers [39], nanosheets, and microspheres [25,40] have been developed for catalysing hydrazine sensing and have shown unique advantages over traditional sensors. In particular, two-dimensional (2D) copper oxide nanosheet architectures have shown a significant hydrazine oxidation activity because of their shortened ion and electron diffusion pathways, an abundance of mesoporous channels and active sites for catalysts, excellent mechanical properties, and improved structural stability [40,41,42].

Several synthesis methods such as hydrothermal routes [43,44], chemical bath deposition methods (CBD) [45], spray pyrolysis [46], microwave irradiation [47], and sol–gel [48] have been reported for the preparation of CuO-NS and their application as sensing electrodes. Considerable focus has been on chemical methods that allow one to control the morphology and particle size, as there is evidence that particle size and morphology directly affect the properties of the catalysts [49,50]. This has been accomplished by considering several synthesis factors such as the type of surfactant template, solvents, temperature, and reactant concentration [40,51]. Recently, the soft-template approach as the chemical method has been widely exploited because of its one-step route that produces high-surface-area mesoporous nanostructured compounds [52,53,54]. Using this approach, our research group reported a foam-surfactant dual template (FSDT) approach to produce mesoporous Ni/Ni(OH)_2_ 2D nanoarchitecture (nanoflakes) via chemical deposition from a hexagonal lyotropic surfactant template using a sodium borohydride reducing agent [55]. The Ni/Ni(OH)_2_ nanoflake catalyst has shown substantial enhancement of the electrochemical nature of methanol and urea oxidation and glucose sensing reactions [51]. Naikoo et al. successfully synthesised macro/mesostructured porous copper oxide via a facile and environmentally benign soft template method by controlling the parameters of shape, dimensionality, and crystallinity. The obtained results revealed that mesoporous copper oxide materials have an excellent catalytic activity towards the oxidation of phenol because of their unique porous nature [56].

To improve the synthesis of CuO nanostructures and further explore their unique properties, the present work, for the first time, demonstrates a novel chemical deposition of 2D CuO nanosheet architecture at room temperature using the versatile foam-surfactant dual templates (FSDT) of self-assembled Brij^®^58 surfactant liquid crystal template and in situ hydrogen foam exfoliation produced by a sodium borohydride (NaBH_4_) reducing agent. A thin layer of Brij^®^58 surfactant mixed with a copper nitrate solution was firstly self-assembled in a Petri dish and an excess of NaBH_4_ solution addition was followed. The copper ions confined in the aqueous domain of the template were reduced by NaBH_4_ and the in situ foam of hydrogen bubbles concurrently exfoliated the deposited CuO nano-layers. The physicochemical characterisation of the CuO nanosheet crystal structure and nanostructured morphology were performed using X-ray diffraction, N_2_ adsorption-desorption isotherms, X-ray photoelectron spectroscopy, and scanning and transmission electron microscopy. The electrochemical hydrazine oxidation and sensing performance at the CuO nanosheet electrode in an alkaline solution was examined by electrochemical and impedance techniques, and evaluated against that of the *bare*-CuO that was deposited without the surfactant and was compared with the state-of-the-art copper-based-catalysts reported thus far.

## 2. Materials and Methods

### 2.1. Materials and Chemicals

Copper (II) nitrate trihydrate Cu(NO_3_)_2_·3H_2_O, the (Brij^®^58) polyethylene glycol octadecyl ether, hydrazine 80%, uric acid (UA), sodium chloride (NaCl), cysteine, sodium borohydride, NaBH_4_, and Nafion (5% in a mixture of water and lower aliphatic alcohols), were purchased from Sigma Aldrich (Dorset, UK). Isopropanol (>99%) was procured from WINLAB (Leicester, UK) and potassium hydroxide (KOH, 85%, pellets) was bought from LOBA Chemie (Mumbai, India). All of the solutions in the experiment were prepared using deionized water obtained from a Milli-Q ultrapure water purification system. Real samples were obtained from Nova Company (Riyadh, KSA) and using tap water from the laboratory.

### 2.2. Synthesis of CuO Nanosheets (CuO-NS)

The CuO nanosheet catalyst was prepared at room temperature (25 °C) using the approach of the foam-surfactant dual template (FSDT) employing the reducing agent of sodium borohydride (NaBH_4_). In the FSDT approach, the copper ions confined in the intestinal aqueous domain of the Brij@58 hexagonal LCT are chemically reduced by the NaBH_4_, which concurrently produces foam of hydrogen bubbles that simultaneously exfoliate and scale off the deposited catalyst layers [51,55]. In brief, as shown in Figure 1, the copper liquid crystal template (Cu-LCT) mixture was prepared by physically adding 5.0 g of a 0.5 M copper (II) nitrate solution, 2.0 g of the melted non-ionic Brij^®^58 surfactant, and 5.0 g of distilled water in a glass beaker. Using an ultrasonic probe, the template mixture was sonicated until a homogenous solution was obtained (15.0 min) and then transferred to a Petri dish. Then, the surfactant to water ratio was adjusted at 40% via evaporation of the excess water in an open atmosphere at 25 °C to form a thin layer of hexagonal liquid crystal copper template. Then, the gel template in the Petri dish was sprayed with an excess of sodium borohydride (1.0 M) reducing agent. After a few seconds, as shown in Figure 1, excessive hydrogen bubble effervescence and foam were formed, and the copper template mixture changed to black, indicating the reduction of copper ions confined in the liquid crystal template.

### 2.3. Characterisation of CuO Nanosheets

The crystal structure and the surface composition of the as-synthesised CuO nanosheets and *bare*-CuO were analysed by X-ray diffraction (XRD, D2 Phaser Bruker (Berlin, Germany) fitted with Cu Kα radiation (λ = 1.5418 Å), and an X-ray photoemission spectroscopy analysis was conducted with an Escalab 250 spectrometer (XPS) from Thermo Fisher Scientific (Oxford, UK). The CuO nanosheets’ and *bare*-CuO catalysts’ morphology and nanoarchitecture were analysed using a scanning (SEM, JSM-7600F; JEOL, Tokyo, Japan) and transmission electron microscope integrated with an electron diffraction (SAED) analysis unit (TEM, JEM 2100F-JEOL, Tokyo, Japan), separately. The specific surface area and the porosity of the CuO nanosheets were determined with a V-Sorb 2800 Porosimetry Analyser (Microtrac Retsch, Haan, Germany) using the Brunauer–Emmett–Teller (BET) technique.

The cyclic voltammetry (CV), chronoamperometry (CA), and impedance (EIS) analysis electrochemical characterisations techniques were performed using the Metrohm AutoLab (PGSTAT302N, Utrecht, Netherlands) Potentiostatic/Galvanostat. A three-electrode electrochemical cell was used for the electrochemical measurements with a platinum sheet (0.5 × 0.5 cm^2^) and Ag/AgCl as a counter and reference electrode, respectively. The working electrode was made of carbon paper (CP, SIGRACET^®^ GDL 24BC, 1.0 cm × 1.0 cm^2^) modified with catalyst ink and the KOH solution as the electrolyte. For the electrochemical characterisation, the ink of the CuO nanosheets was obtained by sonicating 10 mg of the CuO-NS or *bare*-CuO catalyst, 10 µL of Nafion, 0.5 mL of isopropanol, and 0.5 mL of deionized water in a glass vial for 30 min using an ultrasonication probe.

## 3. Results and Discussion

### 3.1. Characterisation of the CuO Nanosheet Structure

The crystal structure and purity of the as-prepared CuO nanosheets were investigated using X-ray diffraction (Cu-Kα radiation). Figure 1a shows the XRD pattern, which reveals the orientation and crystalline nature of the copper oxide nanosheets. The peaks located at 2θ values of 31.90°, 35.21°, 38.5°, 48.53°, 53.29°, 58.07°, 61.25°, 65.90°, 67.82°, 72.17°, and 74.87° were indexed as the (110), (002), (111), (202), (020), (202), (113), (022), (310), (311), and (222) planes, respectively. The above-mentioned diffraction peaks were matched well with the crystalline monoclinic crystal structure of CuO (ICSD:98-006-9757). These results indicate that the synthesised CuO nanosheets were in their pure form and free of any impurities. Moreover, the average size of the CuO nanosheets estimated from the XRD profile and using Debye–Scherrer’s equation was found to be 10.36 nm [27,29]. The XRD of the *bare*-CuO in Appendix A showed similar results, which indicate that the surfactant use did not affect the CuO crystal structure growth.

Figure 1b shows the N_2_ adsorption and desorption isotherms, as well as the pore size distribution curves (inset) of the *bare*-CuO and CuO-NS catalysts. The adsorption–desorption isotherm curve of the CuO-NS catalyst revealed typical type IV isotherms with a hysteresis loop of type H_1_ in the p/p^o^ region of 0.4–1.0, which could be attributed to the N_2_ gas capillary condensation within the mesoporous structure and the slit pores in-between the nanosheets, indicating that the catalysts had a mesoporous architecture [57].

Moreover, the N_2_ isotherm showed that the *bare*-CuO exhibited a very narrow hysteresis loop with a smaller surface area (20.55 m^2^/g), while CuO-NS had wider hysteresis and a higher capillary uptake at a lower pressure and specific surface area (39.42 m^2^/g), thus signifying the presence of higher mesoporosity. In addition, as shown in the inset of Figure 1b, the pore size distribution curve obtained by the BJH method further confirmed the development of many irregular mesopores within the range of 10–70 nm in the case of the CuO-NS catalyst.

The composition and surface functionality of the as-prepared CuO-NSs were further investigated by XPS analysis. The survey spectrum in Appendix A reveals the main elemental composition of the O 1s and Cu 2p peaks, in addition to that of the C 1s originating from the background and/or the surfactant contamination [58,59]. As shown in Figure 1c, the narrow deconvoluted core spectra of Cu 2p showed that the peaks located at 934.9 eV and 954.3 eV could be assigned to Cu 2p_3/2_ and Cu 2p_1/2_, respectively [57,58]. The binding energy separation between the Cu 2p_3/2_ and the Cu 2p_1/2_ peaks was 19.9 eV, which is characteristic of CuO. Meanwhile, the intense satellite features at 943.2 eV and 963.1 eV could be ascribed to the Cu 2p_3/2_ and Cu 2p_1/2_ in CuO, respectively. The deconvoluted core spectra of the O 1s displayed in Figure 1d shows two peaks at 531.2 and 530.1 eV. The low binding energy peak at 530.1 eV could be attributed to the lattice oxygen within the metal oxide (CuO = O, 530.1 eV), whereas the high binding energy peak at 531.2 eV might be assigned to the low coordination surface oxygen species and surface oxygen defects, which could improve the intrinsic catalytic activity of the CuO catalyst [57,58].

The surface morphology of the CuO-NS catalyst was characterised using scanning electron (SEM) and transmission electron microscopy (TEM) to determine its nanoarchitecture morphology. Figure 2a,b shows the SEM images of the as-synthesised nanostructured CuO-NS taken at different magnifications, revealing the morphology of the overlapped and aggregated nanosheets, as well as the interstitial mesoporous channels with various nanostructure sizes that facilitate the mass transport diffusion of the analyte molecules. In contrast, Figure 2c shows the TEM image of CuO-NS, which confirms the 2D nanosheet morphology of the CuO catalyst. The TEM image revealed the presence of bright and dark areas, which corresponded to the individual and stacked nanosheet structures, respectively. The dark areas indicate the regions with stacking between sheets in different crystallographic directions.

The lattice fringe could be seen in the HR-TEM image shown in Figure 2d. The lattice spacing between the adjacent planes was approximately 0.25 nm, corresponding to the distance between the (111) crystal plane of the CuO diffraction pattern (ICSD:98-006-9757). Furthermore, Appendix A shows the SAED pattern of the CuO nanosheets that reveal the presence of the lattice planes of the monoclinic CuO nanosheet crystal structure (ICSD:98-006-9757). The SAED pattern confirmed the good crystallinity of the CuO nanosheets, which was in good agreement with the XRD results. Moreover, for comparison, Appendix A shows the TEM characterisation image of the *bare*-CuO catalyst deposited in the absence of the surfactant. The TEM image shows that the *bare*-CuO had a thick deposit of bulky micro-particles with the absence of a nanosheet morphology.

### 3.2. Electrochemical Characterisation of CuO-NS Electrode

The electrocatalytic activity of CuO-NS in comparison with that of the *bare*-CuO electrodes towards the hydrazine oxidation and sensing was examined using the cyclic voltammetry technique. Figure 3a shows the results of the cyclic voltammetry conducted at a scan rate of 50 mV/s and in the potential range of −0.2 to 0.8 V versus Ag/AgCl of CuO-NS in comparison with those of the *bare*-CuO loaded on the carbon paper electrode in the absence and presence of 20 mM hydrazine dissolved in 1.0 M KOH as a supporting electrolyte. In the absence of hydrazine, a very small current density was observed at the *bare*-CuO electrode (black line) up to a potential of 0.8 V versus Ag/AgCl. In contrast, the CuO-NS electrode in the absence of hydrazine showed significant current density increases around the potential of 0.6 V versus Ag/AgCl (green line), indicating the commencement of the oxygen evolution reaction at the CuO-NS catalyst in the 1.0 M KOH solution. Upon the addition of 20 mM hydrazine to the 1.0 M KOH electrolyte, the CuO-NS electrode exhibited an enormous oxidation peak (blue line) at the onset potential of −0.03 V and the peak potential of 0.35 V versus Ag/AgCl, which could be attributed to the oxidation of the added hydrazine. The oxidation peak current reached 15.60 mA/cm^2^ during the first cycle and stabilized at a slightly lower current of 14.70 mA/cm^2^ during the successive cycles. In contrast, the *bare*-CuO electrode showed a considerably lower electrocatalytic response to the hydrazine oxidation at a more positive potential of 0.55 V versus Ag/AgCl (red line).

The hydrazine oxidation kinetics of the CuO-NS compared with the *bare*-CuO electrode was investigated by Tafel plots obtained by fitting the linear cyclic voltammetry results to the Tafel equation, as shown in Figure 3b. It can be noted that the Tafel slope of the CuO-NS electrode (125 mV/dec) was significantly much lower than in the case of the *bare*-CuO electrode (323 mV/dec), which indicates much faster charge transfer kinetics and that less overpotential is required to oxidized hydrazine when using the CuO-NS electrode.

Electrochemical impedance spectroscopy (EIS) was employed to confirm the facilitation of the electron transfer kinetics at the CuO-NS electrode. Figure 3c displays the Nyquist plots of CuO-NS compared with the *bare*-CuO-NS electrodes in the 1.0 M KOH solution containing 10 mM hydrazine and in the frequency range from 0.01 to 10^5^ Hz at an applied potential of 0.15 V versus Ag/AgCl. The CuO-NS electrode exhibited a smaller diameter semicircle at a low frequency than the *bare*-CuO electrode, indicating that the CuO-NS electrode was considerably less resistive to the charge transfer. The inset of Figure 3c shows the equivalent circuit that fits the corresponding Nyquist diagrams, where R_s_ is the solution resistance and the R_ct_ represents the hydrazine oxidation charge transfer resistance, and Q_1_ is the double electric layer capacitance of the CuO-NS electrode. The Warburg impedance (W) due to the charge transfer at the catalyst/electrolyte interface was observed in the case of the diffusion-controlled reaction. The corresponding impedance parameters obtained from the fitting circuit are reported in Table 1; it can be observed that the CuO-NS electrode showed a considerably lower R_ct_ value (8.17 Ω) than the *bare*-CuO electrode (52.72 Ω), further evidence that the charge transfer occurred considerably faster at the CuO-NS electrode.

The hydrazine oxidation catalytic current was significantly enhanced with the increase in the CuO-NS catalyst loading, and as shown in Figure 3d, the CVs of the CuO-NS catalyst with different loadings ranging from 50 to 300 µg were obtained at 50 mV/s in 20 mM hydrazine dissolved in the 1.0 M KOH solution. As shown in the inset in Figure 3d, the relationship between the hydrazine oxidation peak current density and the catalyst loading exhibited a volcanic plot with a gradual increase in the oxidation peak current as the catalyst loading increased, reaching a maximum oxidation peak current of 14.80 mA/cm^2^ at a catalyst loading of around 250 µg. The loading effect trend could be related to the increase in the available electrochemical active sites of the CuO-NS catalyst with an increase in the catalyst loading. In addition, the decrease in the hydrazine oxidation current above a catalyst loading of >250 µg could be attributed to the retardation of the electron transfer and the active ion diffusion within the thicker catalyst film.

The cyclic voltammetry and impedance results indicate that the CuO-NS electrode showed a significant oxidation performance and faster charge transfer during the electrocatalytic hydrazine oxidation reaction compared with the *bare*-CuO electrode, which could be related to the nanosheet architecture with an enhanced surface area and electron transfer kinetics [60,61,62,63]. Based on recent studies, the mechanism of hydrazine oxidation by the electrocatalytic process significantly depends on the electrode surface nature and the electrolyte solution [60,61,62,63]. In the case of the CuO electrode in an alkaline solution, the mechanism could be proposed to proceed via the irreversible reactions of the hydrazine oxidation at the CuO electrode where the formed intermediate complex of hydrazine in Equation (1) is oxidized at the CuO-NSs surface, producing the N_2_ gas as described by Equations (2) and (3).
N_2_H_4_ + 2OH^−^ → [HO−N_2_H_4_−OH]^2−^
(1)
[HO−N_2_H_4_−OH]^2−^ + 4CuO → 2Cu_2_O + 2H_2_O + 2OH^−^ + N_2_
(2)
2Cu_2_O + 4OH^−^ → 4CuO + 2H_2_O + 4e^−^(3)

The effect of the hydrazine concentration was also investigated using linear sweep voltammetry (LSV) at 50 mV/s, using 250 µg of the CuO-NS electrode, as shown in Figure 4a.

The CV results show that the hydrazine anodic oxidation current at the CuO-NS electrode increased with the increase in hydrazine concentration (from 5.0 to 45 mM) accompanied by the peak potential shift to a higher potential from around 0.250 to 0.500 V versus Ag/AgCl as the hydrazine concentration increased. Moreover, the correlation between the oxidation peak current and the hydrazine concentration showed two regions of the linear range, as shown in Figure 4b. The first region’s concentration ranged from 1.0 to 5.0 mM, and the second region’s concentration ranged from 10.0 to 45.0 mM with an oxidation current sensitivity of 0.76 mA/cm^2^ mM and 1.06 mA/cm^2^ mM, respectively, indicating a high electrocatalytic activity of CuO-NS towards the hydrazine oxidation because of the nanosheet architecture with a high surface area and the presence of a porous network that facilitated the hydrazine diffusion [60,61,62,63]. Moreover, the presence of two dependent linear equations for the correlation between the oxidation peak current and the hydrazine concentration could be related to the variation in the hydrazine/hydroxide ratio and the adsorption competition between hydrazine/hydroxide ions as well as the oxidation mechanism and rate-determining step, in addition to the N_2_ gas formation at the catalytic active sites of the CuO-NS catalyst [60,61,62,63,64,65]. Figure 4c shows the effect of the scan rates of the hydrazine oxidation at the CuO-NS electrode where the CVs were recorded at different potential scan rates from 10 to 400 mV/s in the 1.0 M KOH solution containing 20.0 mM hydrazine. Note that the oxidation peak currents linearly increased against the square root of the potential scan rates (υ^1/2^) with the regression equation (I_peak_ = 1.57x + 4.16 and regression coefficient (R^2^) = 0.999), as shown in Figure 4d. This behaviour was consistent with the hydrazine oxidation at CuO-NS that occurred via the four- electron transfer and mainly via a diffusion-controlled process [60,61,62,63].

To optimize the electrocatalytic sensing activity of the hydrazine oxidation and sensing at the CuO-NS electrode surface, the effects of the applied oxidation potential and the hydroxide electrolyte concentration were investigated using a chronoamperometric technique. Figure 5a shows the chronoamperometry responses of the CuO-NS (250 µg) electrode measured at various applied potentials (from 0.15 to 0.35 V versus Ag/AgCl) to investigate the sensing electroactivity of the CuO-NS electrode towards the successive addition of 1.0 mM hydrazine in the 1.0 M KOH solution and under stirring conditions. In general, the chronoamperometric response of CuO-NS compared with the *bare*-CuO electrodes showed that the oxidation current at all of the applied potentials gradually increased upon the stepwise addition of the hydrazine analyte. The correlation plots between the oxidation current and the hydrazine concentration are shown in Figure 5b, where the oxidation current exhibited a linear response within the studied hydrazine concentration range (from 1.0 to 7.0 mM). The line slopes (hydrazine sensitivity) gradually increased as the applied potential increased and reached a maximum of up to 0.86 mA/cm^2^ mM at the applied potential of 0.30 V versus Ag/AgCl, which was significantly higher than those at the other potentials and approximately eight times higher than those in the case of the *bare*-CuO electrode (slope = 0.10 mA/cm^2^ mM). Therefore, 0.30 V was chosen as the optimum potential in the following experiments for the further quantitative determination of hydrazine.

The electrocatalyst activity of the CuO electrode towards hydrazine oxidation was highly dependent on the OH^−^ concentration under the alkali condition [66], and to obtain the optimal hydroxide concentration, Figure 5c shows the effect of the OH^−^ concentration on the chronoamperometry profiles of CuO-NS (250 µg) electrode upon the addition of different hydrazine concentrations at an applied potential of 0.30 V versus Ag/AgCl. All of the chronoamperometry curves for the different KOH concentrations exhibited a rapid current increase upon the addition of hydrazine, as shown in Figure 5c. The plots of the hydrazine oxidation current as a function of the hydrazine concentration in the 0.1, 0.5, and 1.0 M KOH solution are shown in Figure 5d. It was observed that in the 1.0 M KOH solution that the CuO-NS electrode displayed the maximum hydrazine sensing performance with a higher sensitivity (0.860 mA/cm^2^) and linearity, with a high correlation coefficient than in the 0.1 M (0.358 mA/cm^2^) and 0.5 M (0.594 mA/cm^2^) KOH solutions. Hence, 1.0 M KOH was used as the optimal solution for hydrazine sensing in this study.

The enhanced electrochemical activity of the CuO-NS electrode towards the hydrazine oxidation at various concentrations could be used to achieve high detection sensitivity, relatively wide linear ranges, and a low limit of detection (LOD) of the hydrazine analyte. Figure 6a shows the chronoamperometry response of the CuO-NS compared with *bare*-CuO electrodes at a constant applied potential of 0.30 V and with various injected low concentrations of hydrazine (from 0.025 to 2.5 mM) and under continuous stirring. The CuO-NS electrode exhibited a significantly larger hydrazine oxidation current and shorter time response of less than 3 s compared with the *bare*-CuO electrode, which indicated the highly enhanced electrocatalytic activity and rapid response time of CuO-NS under the lower concentration of hydrazine. In addition, Figure 6b shows that the CuO-NS sensor displayed more than ten times higher hydrazine sensitivity and a better linear relationship (1.47 mA/cm^2^ mM, (R^2^) = 0.998) in the concentration range of 0.25 to 2.5 mM of hydrazine than the *bare*-CuO electrode (0.11 mA/cm^2^ mM, (R^2^) = 0.996). The limit of detection (LOD) could be calculated as follows [66]:LOD = 3 σ/S (4)
where σ is the background current standard deviation before hydrazine addition and S is the slope of the calibration plot of hydrazine. The achieved limit of detection (LOD) of hydrazine at the CuO-NS electrode reached 15 µM, which was significantly lower than that of the different nanostructure electrode materials reported in the literature [60,61,62,63]. Table 2 summarises the hydrazine sensitivity, linear range, and limit of detection (LOD) for the other electrode materials reported in the literature. The CuO-NS sensor showed a significantly higher sensitivity (1.47 mA/cm^2^ mM) and wider linear range from 0.025 to 2.5 mM concentration, which made the CuO-NSs catalyst have the potential to detect a relatively low hydrazine concentration. In addition, the CuO-NS catalyst was easy to fabricate using the one-pot synthesis approach of the foam-surfactant dual template (SFDT) under ambient conditions.

### 3.3. Selectivity and Stability of the CuO-NS Sensor

Selectivity is also an important factor for hydrazine sensing; hence, an interference analysis of the CuO-NS electrode towards various interferences was conducted using chronoamperometry at the applied potential of 0.30 V versus Ag/AgCl, as shown in Figure 7a. The selectivity of the CuO-NS electrode was measured with the addition of a 10 mM solution of each of the common interferences considered, such as uric acid, urea, cysteine, and NaCl. In this context, a solution of 1.0 M KOH was initially used, followed by the successive addition of 1, 2, and 4 mM hydrazine, and then a 10 mM concentration of the interference of uric acid, urea, cysteine, and NaCl were stepwise added to the test solution under continuous magnetic stirring. It was seen that the initial addition of the hydrazine doses led to a substantial current increase, while the stepwise additions of the interferences of uric acid, urea, cysteine, and NaCl produced almost negligible current responses. Moreover, the further stepwise addition of 1, 2, and 4 mM hydrazine produced a current response enhancement similar to that obtained during the first-time addition of hydrazine, implying that the electrode showed excellent selectivity for hydrazine detection. For the hydrazine electrochemical oxidation stability and efficiency study at the CuO-NS electrode, Figure 7b shows the current–time signal response during prolonged oxidation at an applied potential of 0.30 V versus Ag/AgCl and after the successive addition of 5.0 mM and 10 mM hydrazine to 1.0 M KOH under magnetic stirring. The results show that the oxidation current density signal was extremely stable for more than 1.0 h with a negligible current change observed upon the addition of either the low (5.0 mM) or the high (10.0 mM) concentration regime of hydrazine.

Figure 7c shows the long-term stability of the CuO-NS catalyst during storage for 30 days in the air, as investigated every 10 days, by the chronoamperometric response to the hydrazine successive stepwise addition from 1.0 to 7.0 mM in the 1.0 M KOH electrolyte under magnetic stirring and using a 250 µg loading of the CuO-NS catalyst. As shown in Figure 7c, the CuO-NS electrode exhibited a very stable hydrazine oxidation current with almost no change in the hydrazine sensitivity (average sensitivity of 0.87 mA/cm^2^ mM) during frequent usage for 30 days and storage in air, as shown in the inset in Figure 7c, which confirmed the high stability behaviour of the CuO-NS electrode during the sensing of hydrazine.

### 3.4. Electrochemical Analysis of Real Samples

Next, the feasibility of the CuO-NS electrode in a real tap water sample for hydrazine sensing collected from Riyadh and bottled drinking water supplied by Nova Company was evaluated. The chronoamperometric response of the CuO-NS electrode (250 µg loading) towards the hydrazine oxidation was observed with the successive stepwise addition of 1.0 mM hydrazine to 50 mL of the 1.0 M KOH solution containing 5.0 mL from an unknown concentration of the real samples and under magnetic stirring. The recovery experiments were carried out through the standard addition method for the amperometry current response at various concentrations of hydrazine in the real samples. As shown in Appendix A, the recovery results were in the range of 97–108% for tap water and 99–106% for bottled drinking water, indicating that the CuO-NS electrode is a promising electrode for hydrazine determination in real samples.

## 4. Conclusions

In summary, a nanosheet architecture of the copper oxide (CuO-NS) electrocatalyst was obtained at room temperature via a versatile one-pot foam-surfactant dual template (SFDT) synthesis approach using sodium borohydride (NaBH_4_) as a reducing agent. NaBH_4_ chemically reduced the copper ions confined in the aqueous domain of the Brij^®^58 surfactant template, as well as in situ generating excessive hydrogen foam that exfoliate the deposited copper oxide nanosheets. The as-prepared CuO-NS electrode exhibited a remarkably enhanced electrocatalytic activity and sensitivity, low detection limit, and excellent selectivity and stability towards hydrazine oxidation in an alkaline solution. The electrochemical CuO-NS sensing activity for hydrazine oxidation revealed that the CuO nanosheets had a superior sensing performance compared with the bare-CuO and reported state-of-the-art catalysts due to the nanosheet architecture and active sites’ high surface area, which facilitated the charge transfer as well as the mass transport diffusion of the hydrazine molecule. In addition, hydrazine detection in tap and bottled water using CuO-NS electrodes was achieved with a good recovery, which confirmed the potential of CuO-NS electrodes for practical use. The foam-surfactant dual template (SFDT) is a one-pot synthesis approach that could be applied to produce a wide range of nanomaterials with various compositions and nanoarchitectures at room temperature for enhancing electrochemical catalytic reactions.

## Data Availability

Not applicable.

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
