# Peer review of "Hydrazine High-Performance Oxidation and Sensing Using a Copper Oxide Nanosheet Electrocatalyst Prepared via a Foam-Surfactant Dual Template"

_nanomaterials, 2022, doi:10.3390/nano13010129_

Round 1
Reviewer 1 Report
This paper was well written and experimental result and discussion are complete and clear. The topic of this article is very interesting for me. It has a high academic deepth and value. This paper indicated some good findings that the as-prepared CuO-NS electrode showed a remarkably enhanced electrocatalytic activity, sensitivity, low detection limit, and excellent selectivity and stability towards hydrazine oxidation in an alkaline solution. The electrochemical CuO-NS sensing activity for hydrazine oxidation revealed that the CuO nanosheets had superior sensing performance as compared to the bare-CuO and reported state-of-the-art catalysts with a high hydrazine sensitivity of 1.47 mA/cm2 mM, a low detection limit of 15 μM (S/N = 3), and a linear concentration range of up 416 to 45 mM. The foam surfactant dual template (SFDT) is a one-pot synthesis approach that could be applied to produce a wide range of nanomaterials with various compositions and nanoarchitectures at room temperature for enhancing the electrochemical catalytic reactions. Therefore, I recommend this paper can be accepted for publicaiton in Nanomaterials.
Author Response
Many thanks to the reviewer, we appreciate the positive comments and the highlighting of the work findings.
Reviewer 2 Report
Manuscript deals with the Hydrazine high-performance oxidation and sensing using copper oxide nanosheet electrocatalyst prepared via a foam-surfactant dual template. But publication point of view some modifications are necessary.
1) What is the novelty of the present work? Rewrite it at the end of the introduction section.
2) In the case of XRD add a standard stick pattern.
3) To enrich the literature in the introduction section add some references.
i) Surfaces and Interfaces 26, (2021) 101425, ii) Materials Research Bulletin 133, (2021) 111026, iii) Journal of colloid and interface science 576, (2020) 476-485
4) How is SEM morphology useful for this application?
5) Explain in detail the oxidation and sensing mechanism in the revised manuscript.
6) In the conclusion section add important conclusions of the present work not the results.
Author Response
Manuscript deals with the Hydrazine high-performance oxidation and sensing using copper oxide nanosheet electrocatalyst prepared via a foam-surfactant dual template. But publication point of view some modifications are necessary.
1) What is the novelty of the present work? Rewrite it at the end of the introduction section.
Reply: The work reports for the first time the synthesis of a novel mesoporous CuO nanosheet catalyst that was chemically prepared via our developed foam-surfactant dual template synthesis approach using a thin layer of self-assembled liquid crystal template at room temperature. In compliance with this comment, the objective is highlighted throughout the manuscript.
2) In the case of XRD add a standard stick pattern.
Reply: The standard XRD pattern has been added to the experimental one in Fig. 1a.
3) To enrich the literature in the introduction section add some references.
- i) Surfaces and Interfaces 26, (2021) 101425, ii) Materials Research Bulletin 133, (2021) 111026, iii) Journal of colloid and interface science 576, (2020) 476-485
Reply: The above mention references have been noted. They reported the hydrothermal synthesis of copper molybdate, copper oxide, and copper cobaltate respectively for application in electrochemical production and storage. To maintain citing the most relevant references and avoid excessive same authors citation, two of the above references have been cited within the general context of the introduction.
4) How is SEM morphology useful for this application?
Reply: The SEM characterization gives evidence of the existence of the 2D nanosheet morphology as well as the interstitial mesoporous channels of the CuO catalyst at a wider scale.
5) Explain in detail the oxidation and sensing mechanism in the revised manuscript.
Reply: As reported in the literature the mechanism of hydrazine oxidation by the electrocatalytic process is complex and significantly depends on the electrode surface nature and the electrolyte solution. Based on literature reports [ref. 60-65] studies, the mechanism could be proposed for the irreversible reactions of the hydrazine oxidation on the CuO electrode in the alkaline solution. In an alkaline solution, the intermediate complex of hydrazine as shown in Eq. (1) (page 8) is oxidized at CuO-NS surface leading to the N2 as described by Eqs. (2) and (3) are shown on page 8. The text on page 8 is modified in reply to this point.
6) In the conclusion section add important conclusions of the present work, not the results.
Reply: Many thanks for the comment. The conclusion has been reworded and the work findings are highlighted.
Reviewer 3 Report
In this manuscript, authors proposed a method that the nanosheet architecture of copper oxide (CuO-NS) electrocatalyst was prepared via a versatile foam-surfactant dual template (FSDT) approach at room temperature. The copper ions confined in the aqueous domain of the template were reduced by NaBH4, and in-situ generates excessive hydrogen foam and exfoliates the deposited copper oxide nanosheets. The as-prepared CuO-NS electrode exhibited a remarkably enhanced electrocatalytic activity, sensitivity, low detection limit, and excellent selectivity and stability towards hydrazine oxidation in an alkaline solution. The manuscript is organized well, and the data is comprehensive. But there are still some problems to be solved. Therefore, it needs a minor revision before publication to Nanomaterials, as addressed below.
1. The novelty of this work needs to be further emphasized by the authors in introduction.
2. The standard PDF card of CuO should be added in Figure 1a.
3. The inset in Fig.3c should be re-formed.
4. In Figure 4b, for the correlation between the oxidation peak current and the hydrazine concentration of CuO-NS electrode, it seems that there were two dependent linear equations, please provide detailed explanation.
5. There are some irregular writings. Please carefully make corrections throughout the whole manuscript.
(1) Page 10, Figure 5c, “mA/cm” should be “mA/cm2”.
(2) In Fig. 6, line 225, “105” should be “105” (superscript).
(3) “the OH- concentration” should be “the -OH concentration”.
Author Response
In this manuscript, authors proposed a method that the nanosheet architecture of copper oxide (CuO-NS) electrocatalyst was prepared via a versatile foam-surfactant dual template (FSDT) approach at room temperature. The copper ions confined in the aqueous domain of the template were reduced by NaBH4, and in-situ generates excessive hydrogen foam and exfoliates the deposited copper oxide nanosheets. The as-prepared CuO-NS electrode exhibited a remarkably enhanced electrocatalytic activity, sensitivity, low detection limit, and excellent selectivity and stability towards hydrazine oxidation in an alkaline solution. The manuscript is organized well, and the data is comprehensive. But there are still some problems to be solved. Therefore, it needs a minor revision before publication to Nanomaterials, as addressed below.
- The novelty of this work needs to be further emphasized by the authors in the introduction.
Reply: Many thanks, the last paragraph of the introduction has been revised to highlight the work's novelty
- The standard PDF card of CuO should be added in Figure 1a.
Reply: The XRD CuO card number is added to Fig. 1 and in the text.
- The inset in Fig.3c should be re-formed.
Reply: Many thanks for rising this point. The EIS results have been refitted and the corresponding equivalent circuit is shown in the inset in Fig. 3c.
- In Figure 4b, for the correlation between the oxidation peak current and the hydrazine concentration of CuO-NS electrode, it seems that there were two dependent linear equations, please provide a detailed explanation.
Reply: Yes, the reviewer is right, in Fig. 4b there are two different slopes (sensitivity) for hydrazine oxidation depending on the hydrazine concentration range. This could be due to the variation of the hydrazine/hydroxide ratio and the adsorption competition between the hydrazine/hydroxide complex and hydroxide ions as well as the oxidation mechanism and rate-determining step at the CuO electrode as reported in the literature (ref. [60-65]). The text is amended to answer this comment.
- There are some irregular writings. Please carefully make corrections throughout the whole manuscript.
Reply: The whole manuscript has been revised and typing errors are corrected to the best of our knowledge.
(1) Page 10, Figure 5c, “mA/cm” should be “mA/cm2”.
Reply: The y-axis title is corrected.
(2) In Fig. 6, line 225, “105” should be “105” (superscript).
Reply: Thanks for picking up the error, the number is corrected.
(3) “the OH- concentration” should be “the -OH concentration”.
Reply: All the OH- symbols were corrected to OH−
Round 2
Reviewer 2 Report
Revision made by the author satisfactory and the present form of manuscript should be accepted for publication.